# Assessment of Short Rectangular-Tab Actuation of Supersonic Jet Mixing

**Abhash Ranjan** [1]**, Mrinal Kaushik** [1,*] , **Dipankar Deb** [2] , **Vlad Muresan** [3] **and Mihaela Unguresan** [4]

1   Department of Aerospace Engineering, Indian Institute of Technology, Kharagpur 721302, India; abhashranjan896@gmail.com
2   Department of Electrical Engineering, Institute of Infrastructure Technology Research and Management (IITRAM), Ahmedabad 380026, India; dipankardeb@iitram.ac.in
3   Department of Automation, Technical University of Cluj-Napoca, 400641 Cluj-Napoca, Romania; vlad.muresan@aut.utcluj.ro
4   Department of Physics and Chemistry, Technical University of Cluj-Napoca, 400641 Cluj-Napoca, Romania; mihaela.unguresan@chem.utcluj.ro
*   Correspondence: mkaushik@aero.iitkgp.ac.in

**Abstract:** This work explores the extent of jet mixing for a supersonic jet coming out of a Mach 1.8 convergent-divergent nozzle, controlled with two short rectangular vortex-generating actuators located diametrically opposite to each other with an emphasis on numerical methodology. The blockage ratio offered by the tabs is around 0.05. The numerical investigations were carried out by using a commercial computational fluid dynamics (CFD) package and all the simulations were performed by employing steady Reynolds-averaged Navier–Stokes equations and shear-stress transport $k - \omega$ turbulence model on a three-dimensional computational space for more accuracy. The numerical calculations are administered at nozzle pressure ratios (NPRs) of 4, 5, 6, 7 and 8, covering the overexpanded, the correctly expanded and the underexpanded conditions. The centerline pressure decay and the pressure profiles are plotted for both uncontrolled and the controlled jets. Numerical schlieren images are used to capture the barrel shock, the expansion fans and the Mach waves present in the flow field. Mach contours are also delineated at varying NPRs indicating the number of shock cells, their length and the variation of the shock cell structure and strength, to substantiate the prominent findings. The outcomes of this research are observed to be in sensible concurrence with the demonstrated exploratory findings. A reduction in the jet core length of 75% is attained with small vortex-generating actuators, compared to an uncontrolled jet, corresponding to nozzle pressure ratio 5. It was also seen that the controlled jet gets bifurcated downstream of the nozzle exit at a distance of about 5 D, where D is the nozzle exit diameter. Furthermore, it was fascinating to observe that the jet spread increases downstream of the nozzle exit for the controlled jet, as compared to the uncontrolled jet at any given NPR.

**Keywords:** centerline pressure decay; jet mixing; Mach contour; pressure profile; Schlieren flow visualization

## 1. Introduction

The challenge of supersonic jet flow exhibiting augmented mixing is relevant in numerous technological applications, has been an area of multitudinous research for several years and remains a contemporary topic. It has already been established in the open literature that the ideal ratio of large- and small-scale vortical structures are efficacious for enhanced blending. Inducing these bigger and smaller vortical structures into the jet flow can be done either by active methods where a supplementary

power source (such as microjet) is employed to regulate jet behavior or by passive methods where the regulating energy is drawn forthrightly from the flow that needs to be regulated [1]. Large numbers of researchers have reported jet control by using geometrical modifications in the form of actuator tabs of varying geometries and shapes to alter the behavior of shear layer and its stability. A short rectangular vortex-generator, called the tab, is a little projection into the flow that actuates a pair of oppositely rotating vortical structure, thereby eloquently influencing the flow development of jet.

Recalling our concept of vortex dynamics, it is understood that large scale vortical structures are efficient suction creators and have a shorter life span and for the system to be in agreement with the law of conservation of momentum, these large-scale vortices, also known as entertainers, get tattered into smaller eddies. These small-scale vortical eddies have extended life span and they also act as decent carriers of mass and momentum. Identifying a proper proportion of small- and large-scale vortices that ensures best mixing in a field dominated by turbulence is an arduous task as the flow behavior is uncertain. An indirect approach that might be used to quantify jet mixing is to measure the extent of jet centerline decay. Faster descent implies rapid mixing and vice versa [2]. Usually, the decay of centerline pressure or velocity is observed to quantify jet decay.

Several experimental research around jet mixing have been carried out over the past few decades. Hussain (1986) shed some light on the coherent-structure's approach to turbulence and observed that the stretching of longitudinal vortices leads to turbulence [3]. Bradbury and Khadem (1975) were among the pioneers to study low-speed jets by introducing actuator-tabs and they found that intervention of small rectangular-tabs into the jet flow on the perimeter of the had a pronounced impact on the development of jet [4]. Ahuja and Brown (1989) documented that, for a heated round jet issuing from a Mach 1.12 designed nozzle, there could be a reduction in the potential core length of the jet flow from 6 D to under 2 D (where D is the nozzle exit diameter) by employing two diametrically opposite mechanical tabs [5]. Significant abatement in the plume temperature and screech noise was also documented. Further, Samimy et al. (1991) studied the enhancement in jet mixing by actuation of vortices and observed that tabs induced streamwise vortices in the flow field and it significantly augmented the jet spread [6]. Furthermore, variation in the nozzle geometry and tab placement can either increase or decrease the jet spread. Taking these aspects into account, numerous studies have been conducted over the years by employing actuator-tabs of varying geometries at the exit of the nozzle. Zaman et al. (1992) investigated the implications of vortex generators by considering small actuator-tabs and employing them at the end of the nozzle [7]. Delta-tabs were used and it was found that two delta-tabs, spaced 180° apart, absolutely dichotomized the jet. Kaushik and Rathakrishnan (2013, 2015) clearly revealed that the magnitude and potency of vortices actuated from control tabs can be enhanced with geometrical alterations [8,9]. A recent work by Thillaikumar et al. (2020) suggested that the introduction of vortex actuating tabs in the divergent section of the supersonic nozzle changes the structure of the shock cell, thereby reducing the jet core length [10]. From a recent study conducted by Humrutha et al. (2020), it has been observed that the actuation of vortex owing to passive controls or micro vortex generator (MVG) is responsible for accelerating the shock and thus promote mixing [11]. The above-mentioned studies inarguably disclose the fact that tabs with varying configuration over a supersonic nozzle play a key role in manipulating the shape and size of shock cell which in turn promotes jet mixing.

Jets are mostly studied experimentally as the flow behavior is very complex. The complexities arise because of its turbulent nature, entrainment, enhanced agitation in the flow at relatively higher Reynolds number and significant interaction of waves. However, some numerical work has been done by few researchers with the view to comprehend the flow behavior. Launder and Spalding (1974) studied the effectiveness of $\kappa - \epsilon$ model and concluded that it is a simple yet powerful model that grants decent prediction of both near-wall and free-shear- flow phenomena without any kind of modification in constants or functions, but the model is accurate only for a low Reynolds number [12]. Later on, Pope (1978) introduced the effect of correction factor in the standard $\kappa - \epsilon$ turbulence model with a view to anticipate the potential core length with higher accuracy [13]. Tide and Babu (2008) conducted

numerical simulation for a Mach 0.9 compressible jet and observed that the shear stress transport (SST) $\kappa - \omega$ model has a higher accuracy in predicting the mean, turbulence quantities along with acoustic quantities as compared to the Wilcox $\kappa - \omega$ model [14]. Meanwhile, Gross et al. (2010) used an overflow code with different turbulence models to perform turbulence model correction on supersonic jets and observed that, out of all the models employed, the SST model performed exceptionally well [15]. Furthermore, Chin et al. (2013) used both SST $\kappa - \omega$ and $\kappa - \epsilon$ model to predict the flow behavior of supersonic free and impinging jets and later concluded that both the models equally predicted the shock structure in the jet core [16]. Later, Medeiros et al. (2014) compared the large eddy simulation (LES) with unsteady RANS and RANS $\kappa - \omega$ SST turbulence models by performing numerical simulations on supersonic flows and observed that $\kappa - \omega$ SST model was in better concurrence with the experimental results as compared to the other model [17]. Kaushik et al. (2015) also conducted extensive review work to discuss the effectiveness of Reynolds-averaged Navier–Stokes (RANS) model in predicting the supersonic flow behavior of jets. The concluding point revealed that RANS model being an inexpensive and conventional method predicts supersonic core length and decay rate with sufficient accuracy [18]. Furthermore, in a recent study conducted by Jana et al. (2020) for subsonic jets using RANS model, it has been observed that shear layer growth is a prominent and rudimentary phenomenon for mass entrainment [19].

The computational investigations done so far in the area of jet flows encompasses uncontrolled flow behavior with changes in the nozzle cross-section to analyze its effects. The present investigation is an attempt to explore the controlled behavior of the supersonic jet flow computationally as this area has not been studied extensively. A Mach 1.8 convergent-divergent nozzle is considered, and it is regulated by employing two short vortex-actuating rectangular-tabs located diametrically opposite to each other. The tabs offered a blockage ratio of around 5. Simulations were carried out on a three-dimensional computational space by applying steady (RANS) Reynolds-Averaged-Navier–Stokes equations and shear-stress transport $\kappa - \omega$ turbulence model. Rectangular short tab offers two sharp corners and three sharp edges that induce uniform shaped counter rotating vortices responsible for enhanced blending of the jet. The domain is discretized with hexahedral mesh having about 2.7 million cells bubbled out after diligent execution of Grid Independence Test. The maximum skewness for the computational domain came out to 0.48. Simulations were performed with varied levels of nozzle pressure ratios so as to encompass the overexpanded, the correctly expanded and the underexpanded conditions. ANSYS FLUENT, which is a commercial CFD package, was used to carry out this investigation and the three-dimensional computational domain is modelled in Design Modeler which is a designing package that comes inside the ANSYS 16.0 workbench. The descent of centerline pressure along with the variation in the pressure profiles were depicted, for both uncontrolled and the controlled jets. Numerical schlieren images were used to capture the shocks appearing in supersonic jet core, the barrel shock, the expansion fans and the Mach waves extant in the region of the flow field. Mach contours were also depicted at varying NPRs indicating, the number of shock cells, their length and the variation of the shock cell structure and strength, to substantiate the prominent findings. The outcomes of the study were observed to be in accordance with the established experimental findings. It is observed that, the number of prominent shock cells escalates following the escalation in the nozzle pressure ratio (NPR). The aftereffect of introducing tabs results in an increase in the amount of jet spread, as compared to the uncontrolled jet. The results showed jet bifurcation, downstream of the nozzle exit due to augmented mixing. The percentage core length reduction is observed to increase from NPR 4 to NPR 5 and then monotonically decrease with the increase in NPR. This implies that the jet performed better for overexpanded level. An abatement in jet core length of about 75% is achieved corresponding to NPR 5. The jet bisection induced by the short rectangular tabs starts from about 5 D downstream of the nozzle exit.

## 2. Materials and Methods

### 2.1. Modelling and Numerical Domain

The three-dimensional computational model of the nozzle along with the far field region was designed on Ansys Workbench by conscientiously calculating the respective design parameters for the nozzle using the Area–Mach number relation. The nozzle inlet diameter was chosen to be 30 mm in diameter and the throat diameter was about 20 mm, as shown in Figure 1. A divergence angle of 7° was given to the nozzle. An area ratio of about 1.439 corresponding to the Mach number of 1.8 was calculated, which gave a nozzle exit diameter (De) of about 24 mm. The dimensions of the rectangular tab are chosen such that both tabs collectively offer an obstruction of about 5%. From the literature, it is established that tabs are ineffective in the overexpanded flow but in case of under expanded flow, tab length has a profound impact in enhancing jet mixing compared to tab width for same projected area. Increasing the tab length causes the vortical structures to directly interact with the jet core thus increasing jet mixing. Moreover, decreasing the tab width decreases the jet decay. Furthermore, from the literature it is also established that symmetricity leads to rapid jet development, i.e., two tabs placed diametrically opposite to each other will cause complete jet bifurcation. Therefore, this configuration was chosen such that the blockage offered by the tabs should not exceed more that 5% of the nozzle exit area [20].

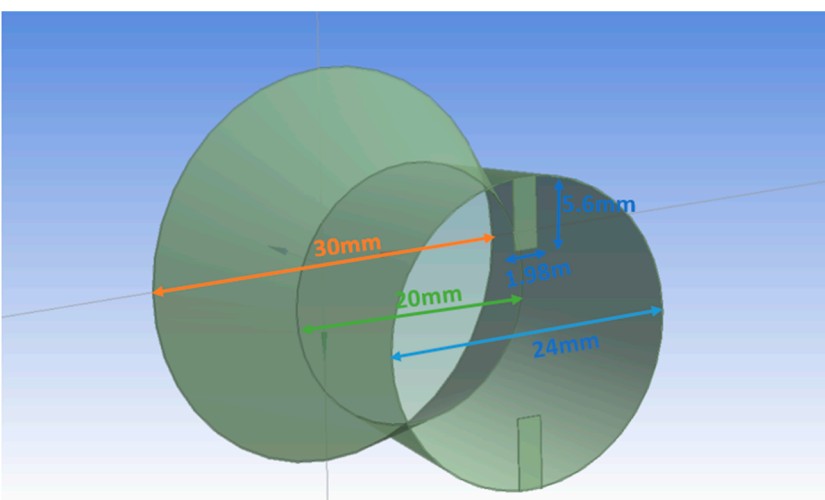

**Figure 1.** Mach 1.8 convergent-divergent nozzle with short rectangular tabs employed at the nozzle exit.

The dimensions of the circular far field domain are chosen in a way to account for capturing the shocks and associated flow properties. The far field stretches to about 30 De in the axial direction because the jet velocity turns out to be inconsequential after about 30 De, even though theoretically it extends up to infinity (Figure 2). The transverse extent of the domain lies to almost 5 times of De, beyond which the flow behavior is insignificant.

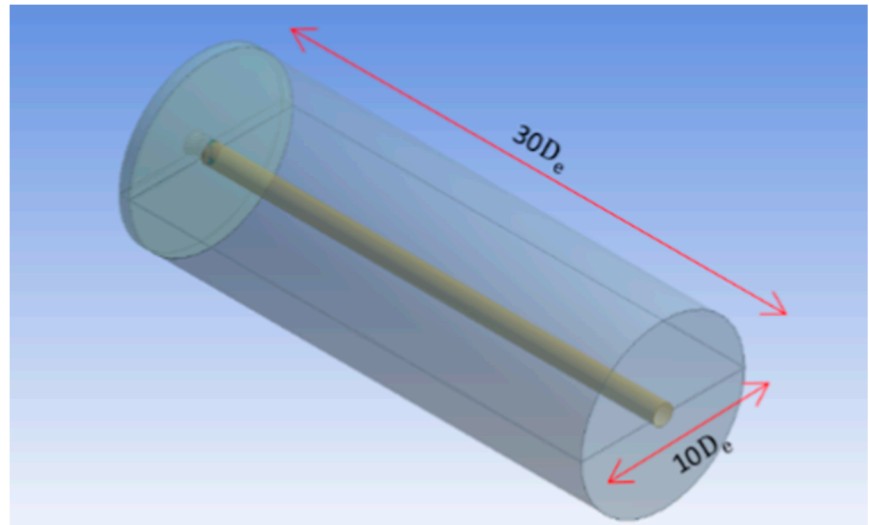

**Figure 2.** Three-dimensional circular domain.

### 2.2. Governing Equations

The conservation form of the governing equations delivers numerical—as well as programming—convenience in the sense that the continuity, momentum and energy equation in its conservation form can be represented by the same generic equation [21]. This helps in simplifying and organizing the logic behind a computer program. Focusing our attention on the compressible flows, the three-dimensional continuity equation can very well be written as:

$$\frac{\partial \rho}{\partial t} + \nabla \cdot (\rho V) = 0 \tag{1}$$

Momentum Equation can be expressed as:
X-component:

$$\frac{\partial (\rho u)}{\partial t} + \nabla \cdot (\rho u V) = -\frac{\partial p}{\partial x} + \frac{\partial \tau_{xx}}{\partial x} + \frac{\partial \tau_{yx}}{\partial y} + \frac{\partial \tau_{zx}}{\partial z} + \rho f_x \tag{2}$$

Y-component:

$$\frac{\partial (\rho v)}{\partial t} + \nabla \cdot (\rho v V) = -\frac{\partial p}{\partial y} + \frac{\partial \tau_{xy}}{\partial x} + \frac{\partial \tau_{yy}}{\partial y} + \frac{\partial \tau_{zy}}{\partial z} + \rho f_y \tag{3}$$

Z-component:

$$\frac{\partial (\rho w)}{\partial t} + \nabla \cdot (\rho w V) = -\frac{\partial p}{\partial z} + \frac{\partial \tau_{xz}}{\partial x} + \frac{\partial \tau_{yz}}{\partial y} + \frac{\partial \tau_{zz}}{\partial z} + \rho f_z \tag{4}$$

Energy equation can be expressed as:

$$\begin{aligned}
\frac{\partial}{\partial t}\left[\rho\left(e + \frac{V^2}{2}\right)\right] \quad &+ \nabla \cdot \left[\rho\left(e + \frac{V^2}{2}\right)V\right] \\
&= \rho\dot{q} + \left[\frac{\partial}{\partial x}\left(k\frac{\partial T}{\partial x}\right) + \frac{\partial}{\partial y}\left(k\frac{\partial T}{\partial y}\right) + \frac{\partial}{\partial z}\left(k\frac{\partial T}{\partial z}\right)\right] \\
&- \left[\frac{\partial (up)}{\partial x} + \frac{\partial (vp)}{\partial y} + \frac{\partial (wp)}{\partial z}\right] \\
&+ \left[\frac{\partial (u\tau_{xx})}{\partial x} + \frac{\partial (u\tau_{yx})}{\partial y} + \frac{\partial (u\tau_{zx})}{\partial z} + \frac{\partial (v\tau_{xy})}{\partial x} + \frac{\partial (v\tau_{yy})}{\partial y} + \frac{\partial (v\tau_{zy})}{\partial z}\right. \\
&\left. + \frac{\partial (w\tau_{xz})}{\partial x} + \frac{\partial (w\tau_{yz})}{\partial y} + \frac{\partial (w\tau_{zz})}{\partial z}\right] + \rho f \cdot V
\end{aligned} \tag{5}$$

### 2.3. Turbulence Model

The turbulence model employed to predict the flow behavior of the supersonic jet flow in this presented investigation is the shear stress transport (SST) $\kappa - \omega$ model where '$\kappa$' is the turbulence kinetic energy and '$\omega$' is the specific rate of dissipation of the kinetic energy due to turbulence into internal thermal energy. This model was elaborated by Menter to conflate the sturdy and precise articulation of the $\kappa - \omega$ model best suited for near-wall region with the free-stream independence of the $\kappa - \epsilon$ model suited for the far field region. To accomplish this advantage, the $\kappa - \epsilon$ model is transformed into a $\kappa - \omega$ formulation. This behavior is achieved by multiplying the standard $\kappa - \omega$ model and the transformed $\kappa - \epsilon$ model by a blending function and then adding them together. The blending function is designed such that it activates the standard $\kappa - \omega$ model near the region close to the wall and $\kappa - \epsilon$ model away from the surface [22]. In short, the SST $\kappa - \omega$ model is more reliable and accurate for flows involving adverse pressure gradient, flows around airfoils, transonic shock waves etc.

### 2.4. Turbulence Equations

The transport equations that are employed in the SST $\kappa - \omega$ model are as follows:
Transport equation for Turbulent Kinetic Energy

$$\frac{\partial}{\partial t}(\rho\kappa) + \frac{\partial}{\partial x_i}(\rho\kappa u_i) = \frac{\partial}{\partial x_j}\left(\alpha_k \frac{\partial \kappa}{\partial x_j}\right) + G_\kappa - T_\kappa + S_\kappa \tag{6}$$

Transport equation for Turbulent Kinetic Energy Dissipation

$$\frac{\partial}{\partial t}(\rho\omega) + \frac{\partial}{\partial x_i}(\rho\omega u_i) = \frac{\partial}{\partial x_j}\left(\alpha_\omega \frac{\partial \omega}{\partial x_j}\right) + G_\omega - T_\omega + D_\omega + S_\omega \tag{7}$$

where, $G_\kappa$ represents the generation of kinetic energy resulting from turbulence owing to mean velocity gradients. $G_\omega$, characterizes the generation of dissipation rate. $T_\kappa$, and $T_\omega$ represents the dissipation of $\kappa$ and $\omega$ due to turbulence. $D_\omega$, denotes the cross-diffusion term. $S_\kappa$, and $S_\omega$ are the source terms.

### 2.5. Meshing

The meshing that has been adopted for this present investigation is structured in behavior. The elementary geometry is hexahedral and the overall skewness comes down to about 0.48 which ensures that the continuous geometrical domain contains cells of good shapes. The y+ value for the near-wall mesh came out to be in the range from 5 to 30. There was a variation of about 5 to 10 $\mu$m in the near-wall spacing corresponding to the wall y+ value. Figure 3 shows the meshing of the computational domain and from this figure it can be clearly seen that the region near the jet flow field has been densely meshed so that the shocks can be recorded accurately. The region near the nozzle area is also finely meshed because the mesh should always be fine for areas that are consequential for subsequent computation. Figure 4 shows the magnified lookout of the nozzle region that has been meshed finely.

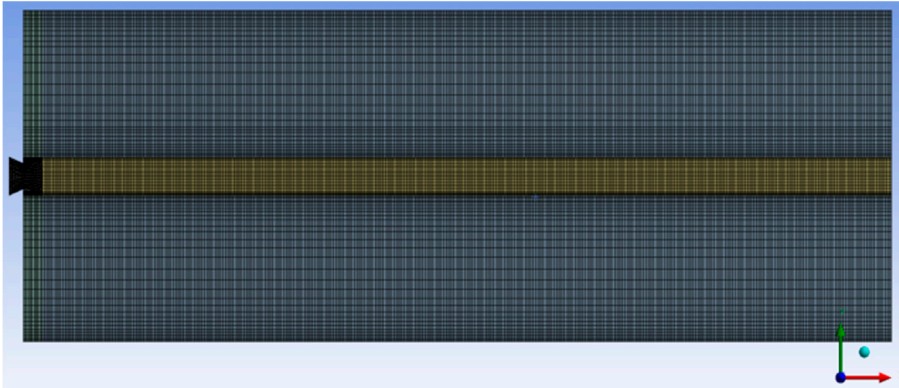

**Figure 3.** Hexahedral meshing of the computational domain.

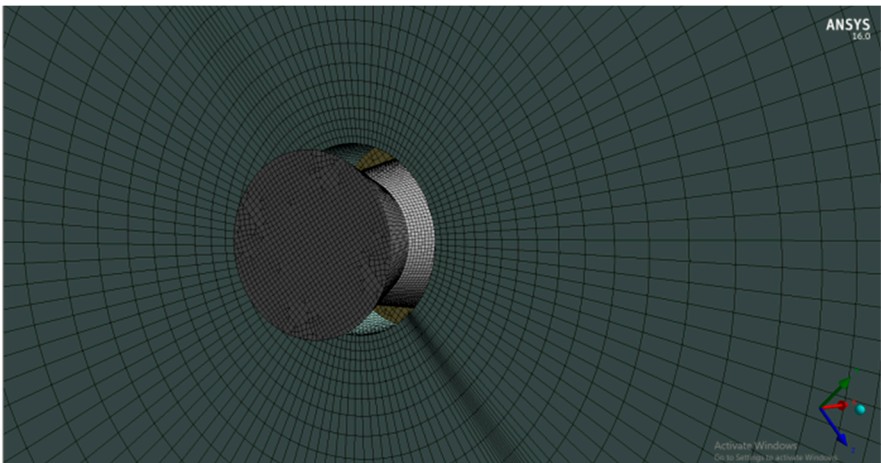

**Figure 4.** Enlarged view of finely meshed nozzle region.

### 2.6. Boundary Conditions

The domain for computation signifying the boundary types and boundary limits are shown from Figures 5–8, respectively. The nozzle inlet is specified as pressure inlet and different pressure values are allocated to it based on the level of expansion desired. The nozzle wall and rectangular tabs are considered as a wall to assimilate the consequence of wall in the flow field. This allows the effect of the tabs to be involved in the calculation. The outer domain known as the far field is modeled as pressure inlet and is given atmospheric pressure as the input. Lastly, the outlet of the far field is given the condition of pressure outlet which is kept at the atmospheric pressure. Supersonic jet flows are highly compressible which means the density variation is highly significant. That is why the density-based solver is adopted in carrying out the calculations. Ideal gas is chosen as the compressible fluid and Sutherland viscosity is specified to it.

### 2.7. Grid Independence Test

The numerical simulations in the present investigation have been performed on grid sizes of 1.1, 1.8, 2.7 and 3.7 million, respectively, to ascertain the optimum grid resolution that will deliver certifiable results. From the Figure 9, it is assuredly observed that as the grid size is augmented from 1.13 million to 1.8 million, the results improve in quality. The same trend is observed when the grid size is further augmented from 1.8 million to 2.7 million. However, a further augmentation of grid size from 2.7 million to 3.7 million fetches no refinement in the centerline pressure decay curve as can be corroborated from the figure below. The CPD of 2.7 million and 3.7 million overlap each other suggesting an optimum grid resolution being reached. Thus, the grid size of 2.7 million is selected for further investigations and numerical simulations.

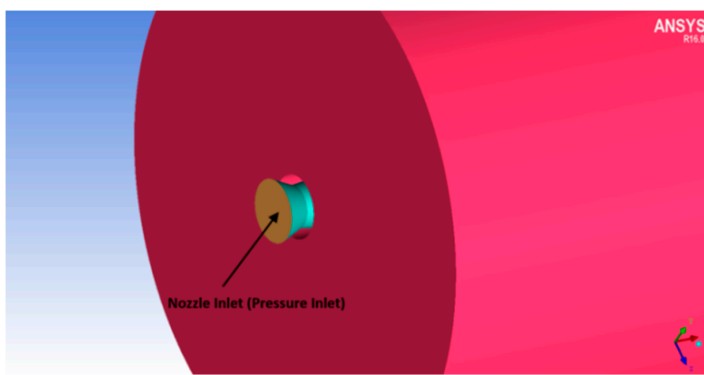

**Figure 5.** Nozzle inlet as pressure inlet.

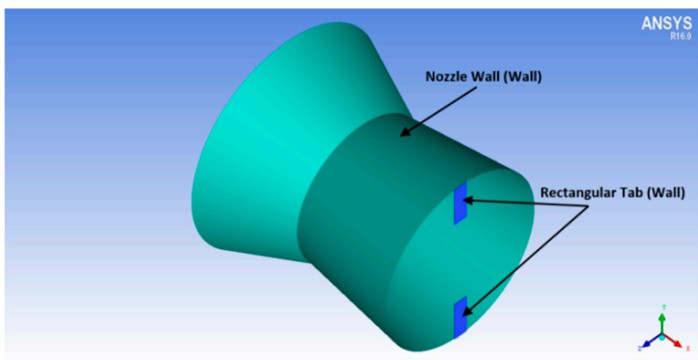

**Figure 6.** Nozzle wall and rectangular Tabs as wall.

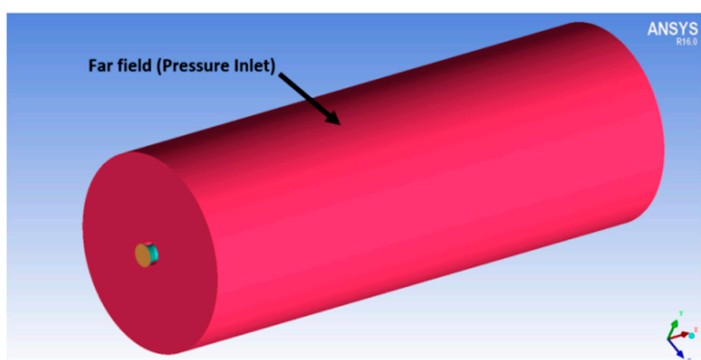

**Figure 7.** Far-field as pressure inlet.

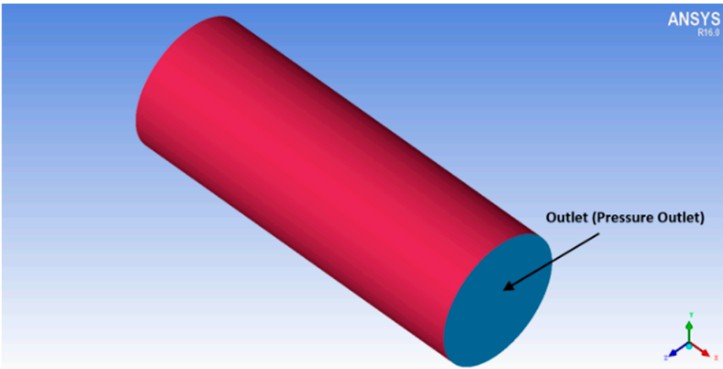

**Figure 8.** Domain outlet as pressure outlet.

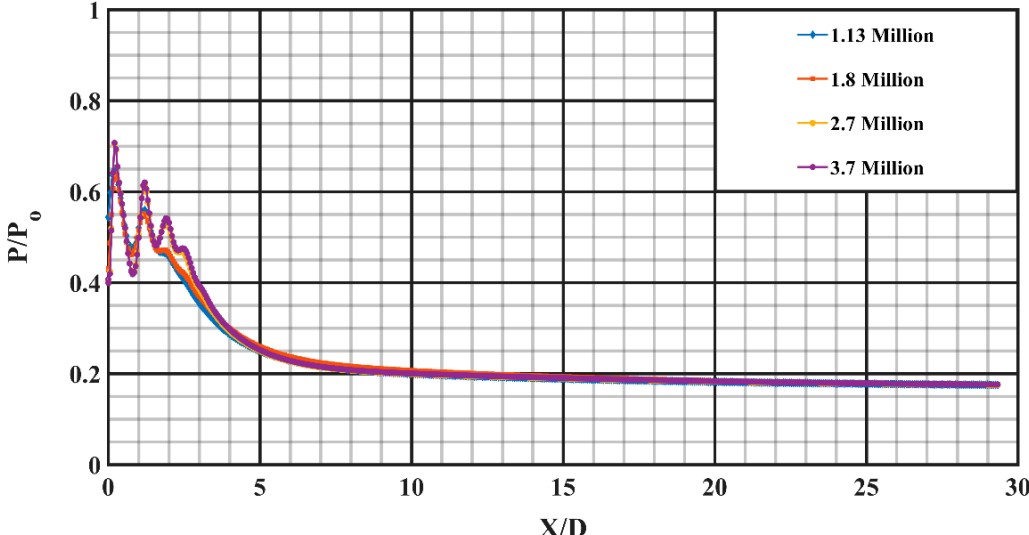

**Figure 9.** Grid independence test (centerline pressure decay to compare the grid resolution).

## *2.8. Validation*

The most important aspect of any computer simulated model is to confirm its accuracy and credibility. Computer simulated models are approximate replications of the real-world problems and they can never imitate the exact problem therefore it should render the exact condition up to a certain acceptable degree. From Figure 10, it can be inferred that the simulated result is in reasonable accordance with the experimental observations [23]. Although there is a slight variation in Mach number, the expansion conditions are exactly same, and for moderate supersonic flows, a slight variation in Mach number does not impact the flow significantly, because the effects of expansion conditions on supersonic core are more prominent than the marginal difference in Mach number. After comparing the computational results with the experimental data, the computational results came out to be fairly accurate. In addition to this, the flow physics is also consistent with the experimental data. Furthermore, these results are also in line with the experimental work done by Chiranjeevi and Rathakrishnan [24].

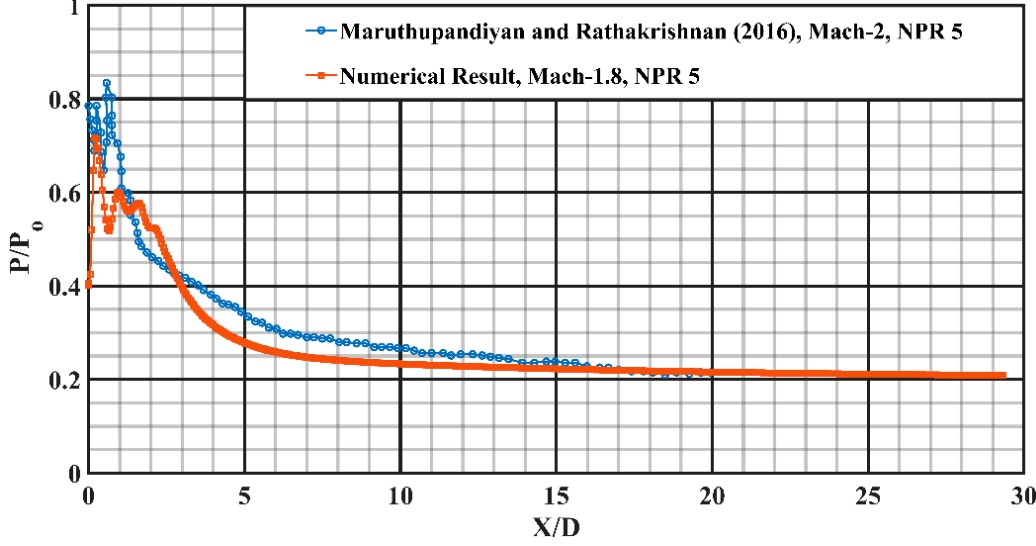

**Figure 10.** Validation of simulated finding with experimental work.

## 3. Results

Supersonic flow is largely subdued by waves, and precise measurement of actual total pressure and Mach number at every spatial position in the direction of nozzle exit is difficult, experimentally or numerically. Consequently, to evaluate the extent of jet blending, the usual practice is to plot the non-dimensional pressure against the non-dimensional axial lengths (X, Y and Z). For validation of these numerical results with established experimental findings, a similar approach is adopted.

### 3.1. Centerline Pressure Decay

Centerline pressure decay is an immediate quantification of jet propagation and mixing that occurs in the jet flow field. A faster decay alludes speedy mixing of the jet fluid mass with the entrained fluid mass. The decay of centerline pressure can most certainly be used to flaunt the confines of the jet core, which is mostly outlined as the axial spread up to which the supersonic jet predominates.

The descent in centerline pressure for both uncontrolled and controlled jets, at the overexpanded condition with an overexpansion level of about 30.43% corresponding to NPR 4, is depicted in Figure 11. In the illustration for an uncontrolled jet, six conspicuous shock-cell structures having significant potency can be clearly observed. The supersonic jet core length elongates up to about X/D = 8. However, with the introduction of rectangular tabs at the end of the nozzle, the jet core length is abated drastically to around X/D = 2, offering a core length abatement of around 74.4%. It is lucidly evident that uniform-sized small-scale vortical structures insinuated into the jet field resulted in enhanced mixing. Furthermore, it is also confirmed that beyond X/D = 20, uncontrolled jets attain self-similar profile while the controlled jet attains it beyond X/D = 10 suggesting the dominancy of early viscous effects owing to enhanced mixing in a controlled jet.

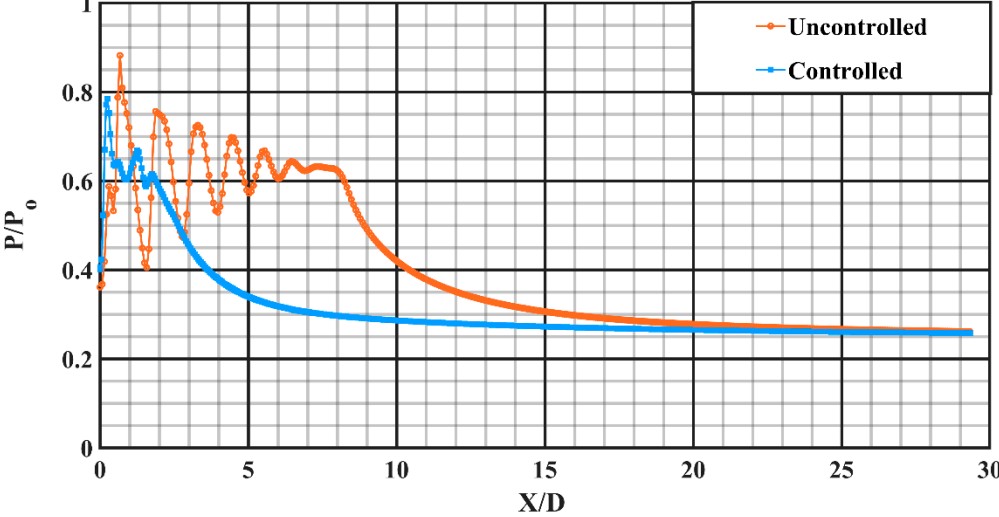

**Figure 11.** Decay of Jet centerline pressure for Mach 1.8 at NPR 4.

At NPR 5, with an overexpansion level of about 13.04%, the centerline pressure decay results for the uncontrolled and controlled jet are shown in Figure 12. The core length for uncontrolled jet for this NPR elongates up to about X/D = 8.84, meanwhile the same for the controlled jet is about X/D = 2.2, offering an abatement in core length of around 75.09%, suggesting that rectangular tabs perform better at the overexpanded condition with adverse pressure gradient at the nozzle end. A trend similar to NPR 4 in the attainment of self-similar profile for the uncontrolled and controlled jet suggests early viscous action. Figure 13 depicts the descent in centerline pressure at NPR 6, which is a case of almost correctly expanded state with a marginal underexpansion level of about 4.35%. The core length for uncontrolled jet elongates up to around X/D = 9.67. Meanwhile, for a controlled jet, it is about X/D = 2.55, offering an abatement in the core length of around 73.60%. The adverse pressure

gradient is still intact with the flow at the nozzle exit; however, the intensity is reduced remarkably as opposed to that at NPR 4.

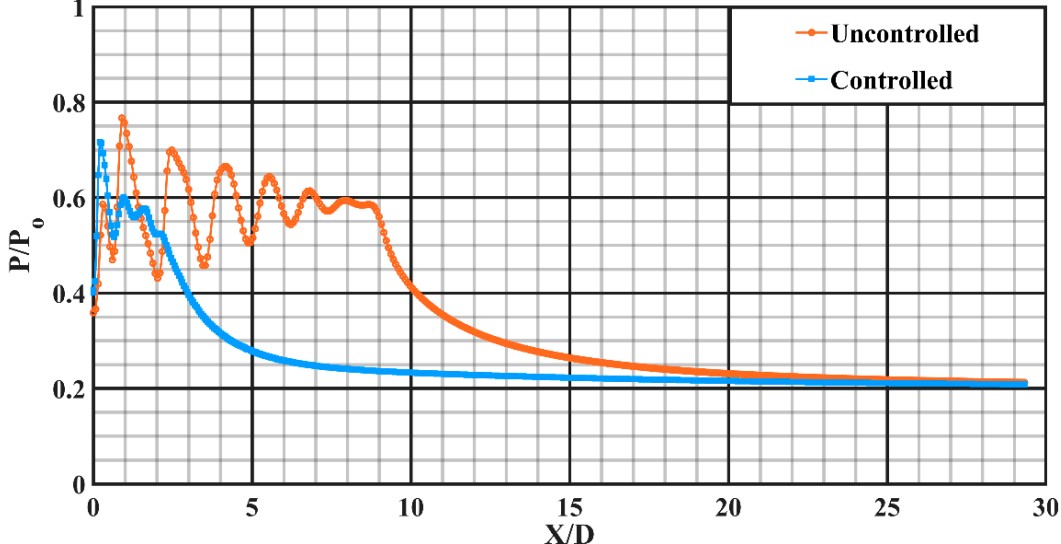

**Figure 12.** Decay of Jet centerline pressure for Mach 1.8 at NPR 5.

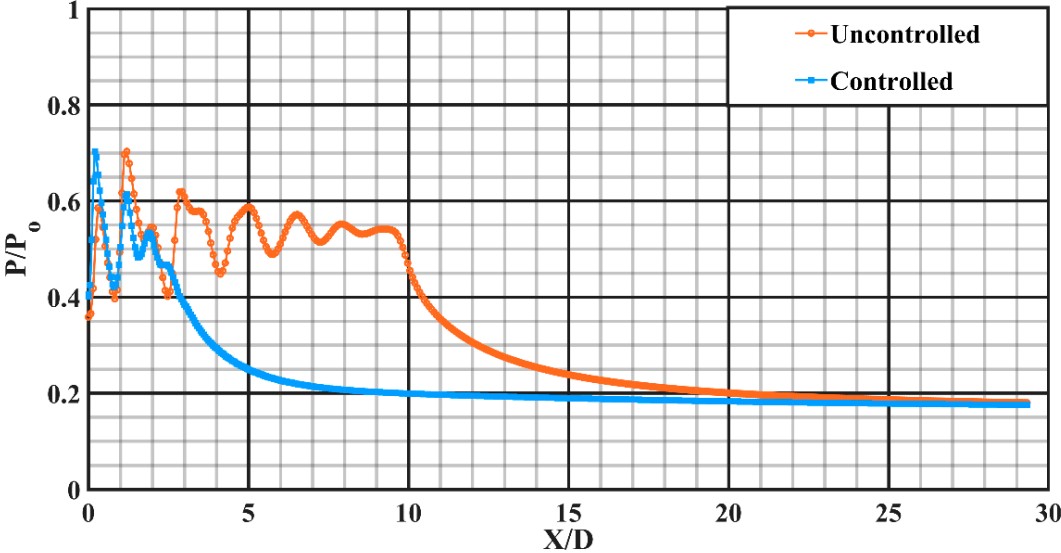

**Figure 13.** Decay of Jet centerline pressure for Mach 1.8 at NPR 6.

At an underexpansion level of about 21.74% corresponding to NPR 7, the results are depicted in Figure 14. It is observed that the supersonic core length for uncontrolled jet elongates up to around X/D = 10.27, although for controlled jet, it outstretches to about X/D = 3.5, that is, core length abatement of about 65.90% is observed for the controlled jet featured with favorable pressure gradient at the nozzle end.

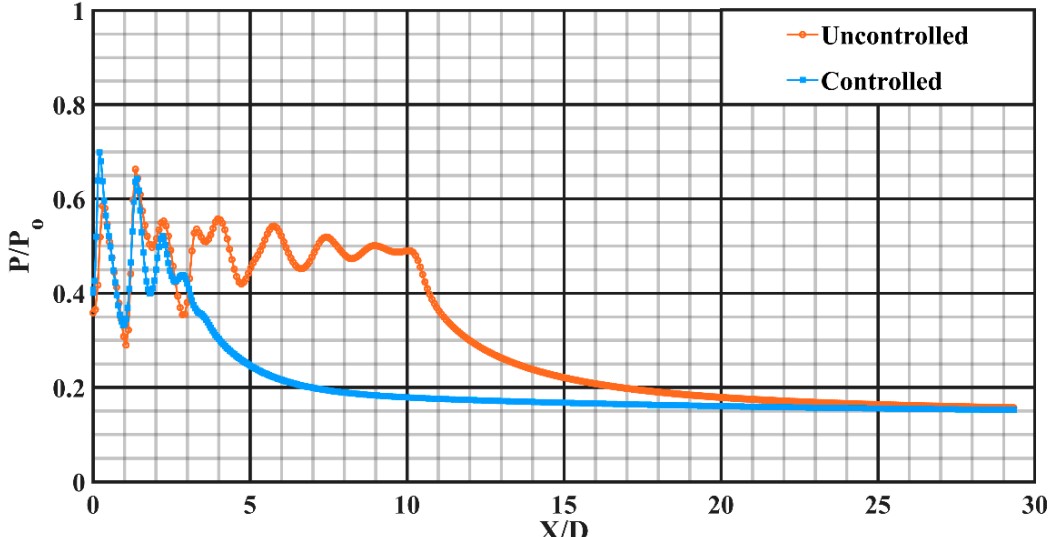

**Figure 14.** Decay of Jet centerline pressure for Mach 1.8 at NPR 7.

At NPR 8, with an underexpansion level of about 27.18%, the centerline pressure decay for uncontrolled and controlled jet is shown in Figure 15. Favorable pressure gradient influences the flow behavior profoundly compared to that at NPR 7 causing the jet field to spread more. It is observed that the jet core length associated with uncontrolled jet stretches to around X/D = 10.65, whereas for a controlled jet, it outstretches to around X/D = 3.9, offering an abatement in jet core length as high as 63.34%.

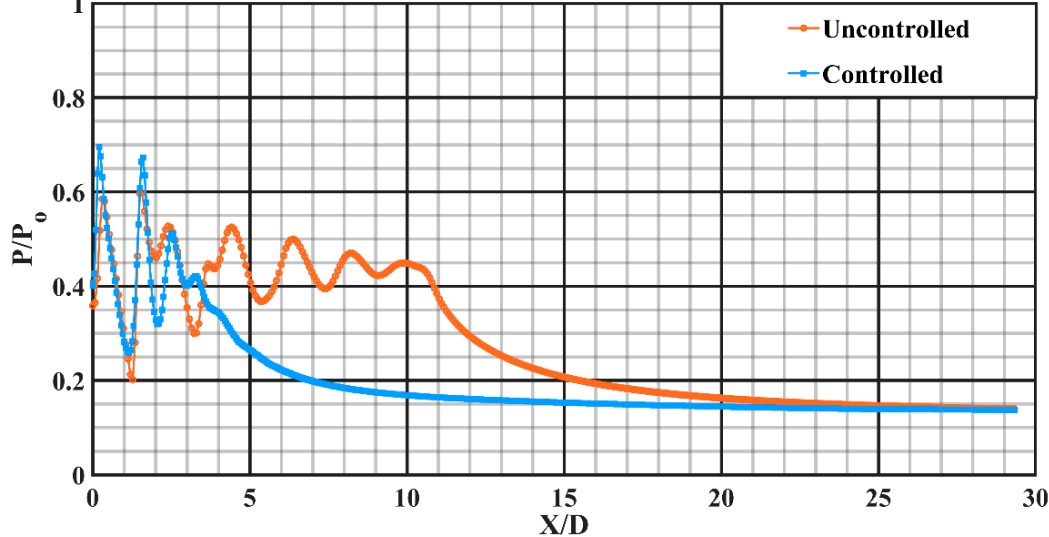

**Figure 15.** Decay of Jet centerline pressure for Mach 1.8 at NPR 8.

From the descent in centerline pressure, it is observable that the efficiency of promoting augmented mixing by rectangular tabs employed at the nozzle end for a Mach 1.8 circular nozzle attains a maximum in the overexpanded condition, implying as high as 75% abatement in jet core length at NPR 5. Rectangular tabs perform better in the overexpanded state by shedding uniform-sized vortices in the vicinity of an adverse pressure gradient. Each sharp corner is accompanied by two opposing vortical structures which interact with each other and induce violent mixing in the region of the jet field from the nozzle end. The numerical quantification of the effectiveness of rectangular tabs is the

rate of the change in core length (from uncontrolled to controlled) to the core length of the uncontrolled jet, expressed as:

$$\text{Percentage Reduction in core length }(\Delta L) = \frac{L_{\text{uncontrolled jet}} - L_{\text{controlled jet}}}{L_{\text{uncontrolled jet}}} \times 100$$

In this presented research, the percentage abatement in core length is calculated for all the NPRs to unravel the optimum condition that results in maximum mixing. The results obtained are tabulated in Table 1, and plotted in Figure 16.

**Table 1.** Core lengths for uncontrolled and controlled jet at different NPRs along with the percentage reduction in core length.

| NPR | $L_{\text{uncontrolled}}$ | $L_{\text{controlled}}$ | % Reduction in Core Length |
|-----|---------------------------|-------------------------|----------------------------|
| 4   | 8 D                       | 2.1 D                   | 74.40                      |
| 5   | 8.8 D                     | 2.2 D                   | 75.09                      |
| 6   | 9.7 D                     | 2.6 D                   | 73.60                      |
| 7   | 10.3 D                    | 3.5 D                   | 65.90                      |
| 8   | 10.6 D                    | 3.9 D                   | 63.34                      |

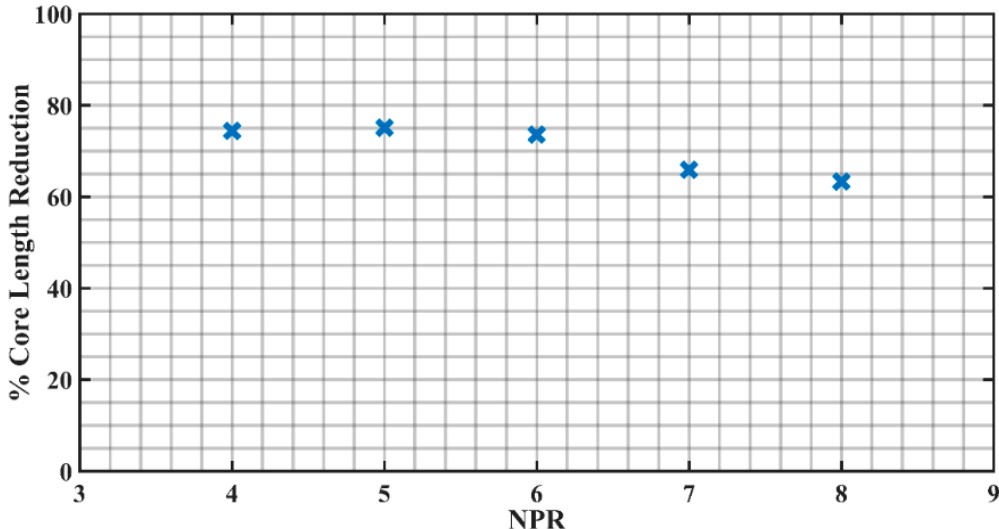

**Figure 16.** Fluctuation in percentage core length reduction with NPR.

Figure 16 shows the fluctuation of percentage core length reduction at different NPRs, and the maximum reduction is observable at NPR 5 which is 75.09%. Moreover, the relational dependency between two adjacent points in the above graph cannot be precisely defined and hence not joined by continuous lines.

## 3.2. Pressure Profiles

A pressure profile depicts the pressure distribution and its variation along any specific direction that indispensably corroborate some aspects of the jet behavior like asymmetry, jet bifurcation, jet spread and so on. The fierce vortex exertion in the nearby field causes the jet to grow unsteady and oscillatory, consequently the pressure profile becomes imperative in visualizing these aspects quantitatively. In this present investigation, pressure distribution is numerically calculated along tab-orientation (Y direction) and perpendicular to tab-orientation (Z direction) for both controlled and uncontrolled jets which is further non-dimensionalized by dividing the numerically calculated total pressure with the pressure given at the inlet of the nozzle. Furthermore, this non-dimensionalized pressure is plotted against the non-dimensionalized distance (which is obtained by dividing the axial distances by the

nozzle end diameter) to capture the pressure distribution profiles at varied NPRs. Figure 17 assists in better comprehending the directions along which the pressure profiles have been obtained.

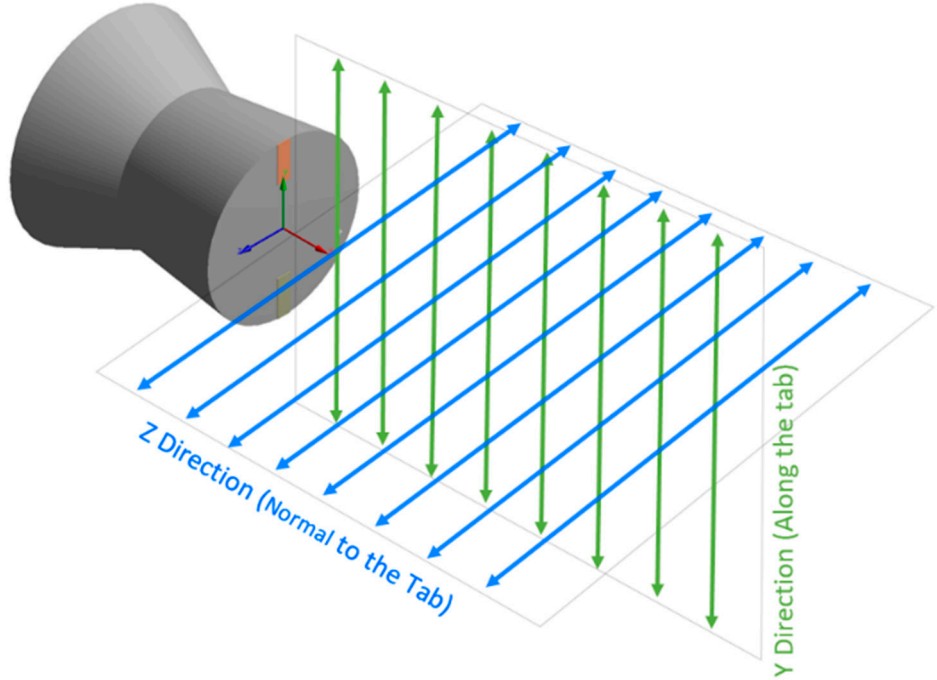

**Figure 17.** Figure depicting directions along and perpendicular to tab-orientation.

Figure 18 shows the pressure profiles for unregulated jet at NPR 5 with an overexpansion level of about 13.04% at varying X/D values. At X/D = 0, it is observed that the pressure profile almost remains constant. It can be speculated that this constant behavior is because of the reflection and compression of fluid particles by the oblique shocks formed at the end of nozzle in an attempt to raise the exit pressure of nozzle up to the back pressure. As the jet progresses and encounters the large free space along the direction of the nozzle end, the pressure profile starts to shift downwards and contracts in proportion around the jet axis owing to the viscous action of the surrounding large space. At X/D = 0.5, the pressure is minimum at the jet axis suggesting maximum velocity corresponding to that point. Further moving slightly away from the axis of jet, the pressure remains almost invariable till a radial distance of 0.4 and then sharply decreases up to 0.6 thereby attaining a constant pressure zone with a value of P/Po = 0.2 depicting a region a constant Mach number. At X/D = 1, there is a decrease in the pressure at the jet axis and it is observed that for X/D = 1, 2, 3, 5 and 8 the pressure difference is marginal and the effect of surrounding viscosity is not very significant as the jet strength is pronounced but as we move beyond X/D = 8, a significant effect of viscosity can be observed. At X/D = 10, the jet enters in the characteristics decay zone, thereafter, attains a self-similar profile.

Figure 19a,b depict the pressure profiles for the controlled jet in the two directions, that is, along tab-orientation (Y-direction) and perpendicular to tab-orientation (Z-direction) for varying X/D values at the corresponding NPR. It is observed that the y-profiles (Figure 19a) exhibits a constant pressure zone in the nearby field zone signifying mixing owing to vortex shedding by rectangular tabs. Further, it does not exhibit any fall of pressure as compared to that for uncontrolled jet in the nearby field region. The jet (y-profile) yields a slightly asymmetric behavior, but this behavior is not pronounced as the computational model is designed with exact dimensions with no errors whatsoever. Furthermore, it is also observed that the peak pressure variation about the jet axis in the nearby region is marginal for X/D = 0.5 and 1.

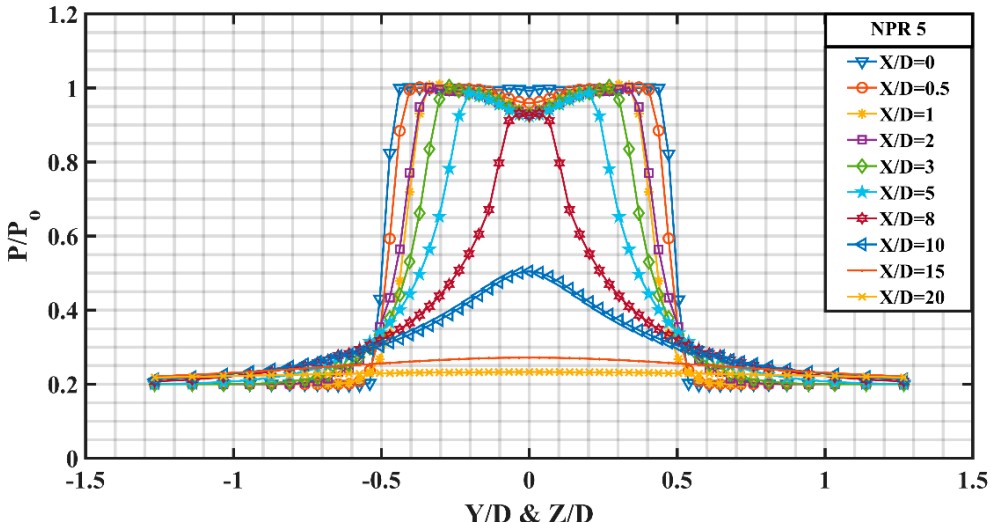

**Figure 18.** Pressure profile variation for uncontrolled jet at NPR 5.

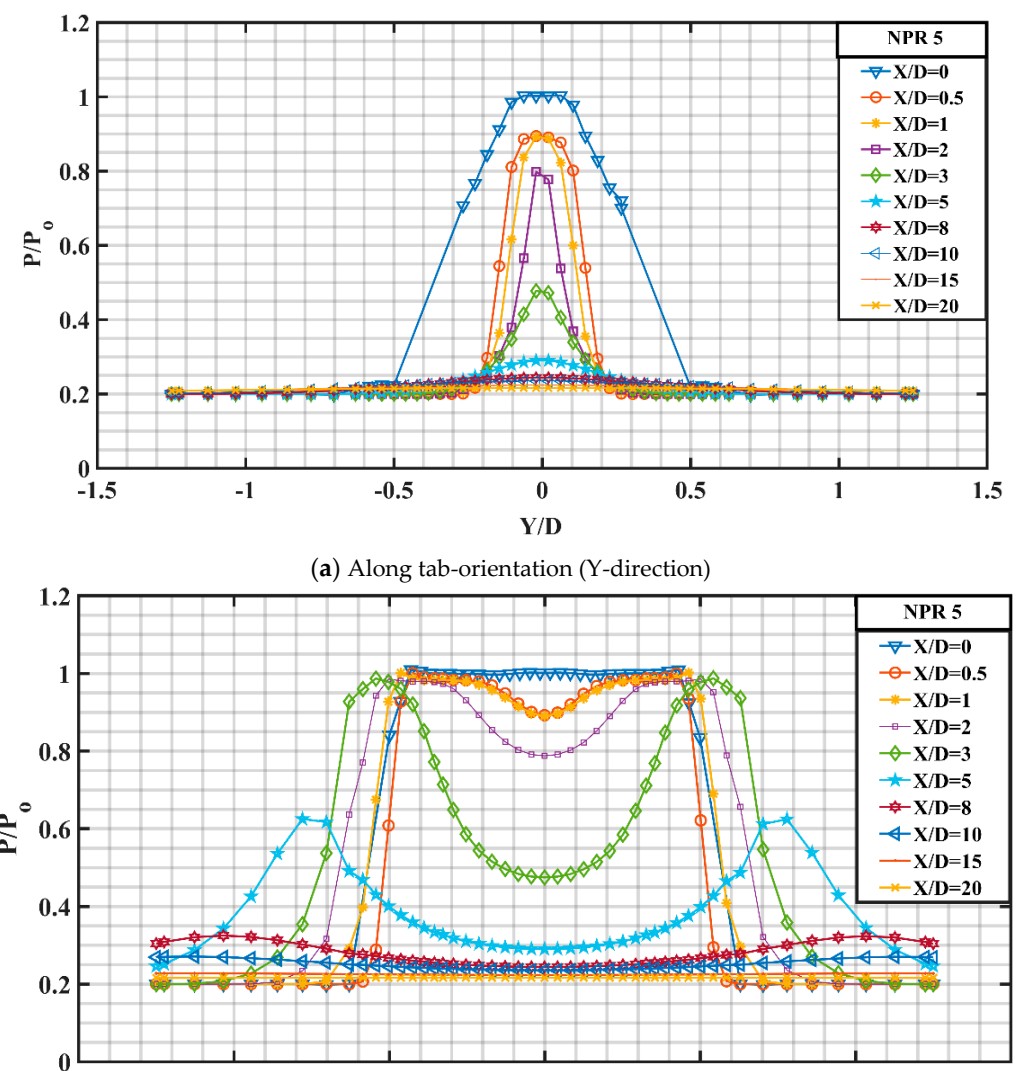

(**a**) Along tab-orientation (Y-direction)

(**b**) Perpendicular to tab-orientation (Z-direction)

**Figure 19.** Pressure profile variation for controlled jet at NPR 5.

Peak pressure along the jet axis decreases downstream of the nozzle exit and in the far field region, the y-pressure profiles shows asymptotic decay appreciably indistinguishable compared to the uncontrolled jet. The variation of pressure profiles in the z-direction (Figure 19b) are disparate from the y-profiles and the jet bifurcation can be observed and substantiated by two off-centered peaks which abate as we progress away from the nozzle end. At X/D = 15 and beyond, the peak flattens out and in addition to this, both the y-profiles and the z-profiles look almost identical to each other and the uncontrolled jet in the far field region.

Figure 20 delineates the variation in pressure profiles for almost correctly expanded jet at Mach 1.8 with a marginal overexpansion level of about 4.6% corresponding to NPR 6. Fairly negligible pressure gradient is offered at the nozzle end and the pressure profile pattern looks somewhat identical to NPR 5. The pressure drop along the jet axis corresponding to X/D = 0.5 indicates supersonic acceleration and the viscous action is not dominant around the jet centerline but because of the boundary layer and its associated viscous action, a constant pressure zone is created away from the jet centerline which further falls sharply and finally attains a constant value of P/Po = 0.18 as the jet approaches self-similar condition. Marginal pressure variation is observed for X/D = 1, 2, 3, 5 and 8. At X/D = 8, a small invariable pressure region around the jet axis is observed. At X/D = 10, the peak pressure is observed to be somewhat dominant than that at NPR 6. The jet finally achieves self-similar behavior in the far field region suggesting fully developed jet.

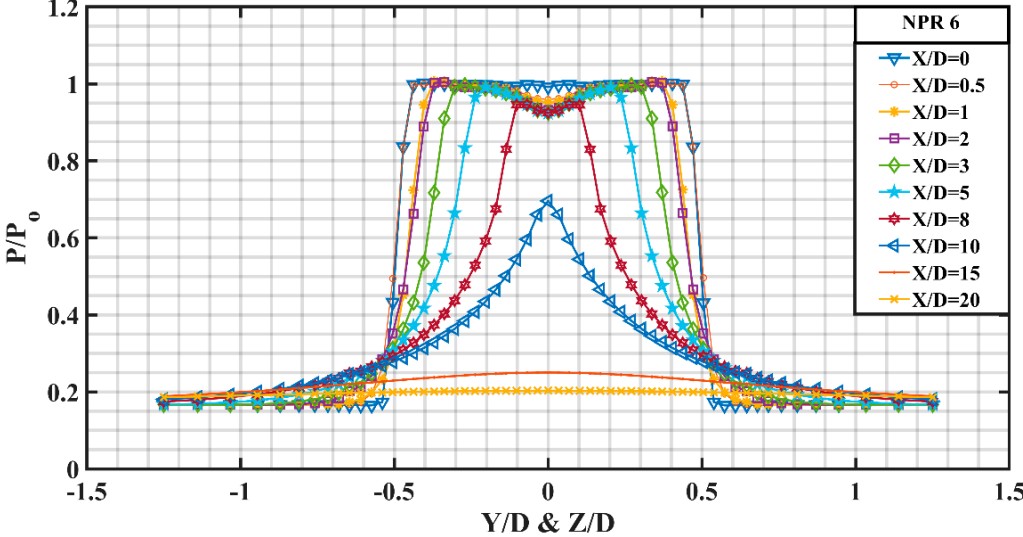

**Figure 20.** Pressure profile variation for uncontrolled jet at NPR 6.

Figure 21a,b depict the y profile and the z profile associated with controlled jet corresponding to NPR 6. Referring to the y-profile, it can be observed that at X/D = 0.5, the peak pressure is around the jet axis, followed by a constant pressure zone around it which is disparate from the pressure profile at NPR 5. Finally, there is a sharp drop in pressure to attain self-similar zone. Shocks of marginal strength can be observed in the near field region. At X/D = 2, 3 and 5, the peak pressure seems to be slightly higher than the corresponding values at NPR 5, owing to higher jet strength. Furthermore, the y-profile attains fully developed behavior downstream of the nozzle exit.

The z-profile for controlled jet at NPR 6 looks somewhat similar to that at NPR 5 with slight variation in the peak pressure. It can be observed that there is a slight escalation in the pressure difference between the profiles at X/D = 0.5 and X/D = 1 comparative to that at NPR 5. At X/D = 3, the two off-centered peaks exhibit a constant pressure zone comparative to that at NPR 5 suggesting better development of the two high velocity fields generated because of the rectangular tabs. No significant change in the far filed region can be observed for the two NPRs.

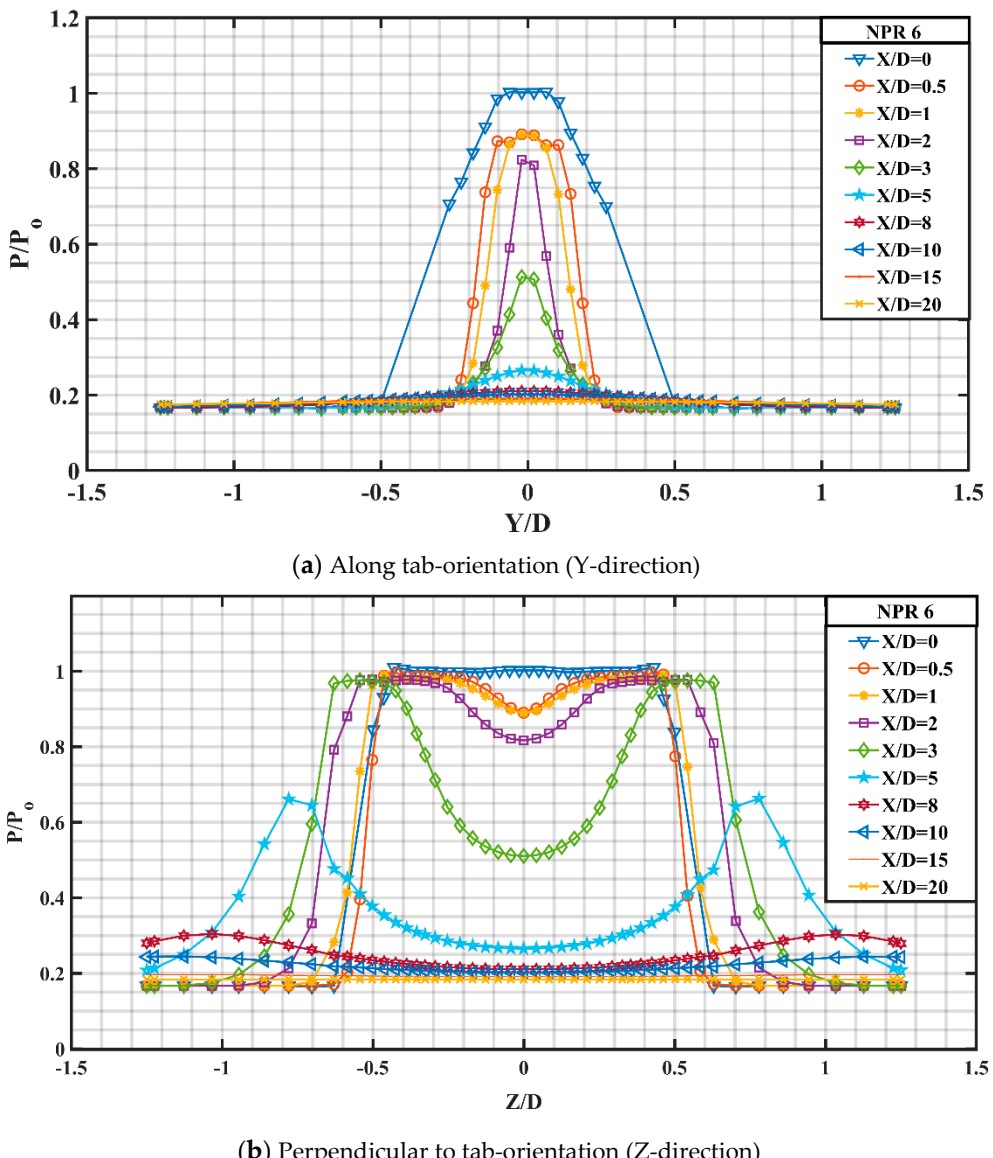

(**a**) Along tab-orientation (Y-direction)

(**b**) Perpendicular to tab-orientation (Z-direction)

**Figure 21.** Pressure profile variation for controlled jet at NPR 6.

Figure 22 shows the variation in pressure profiles for an uncontrolled jet at the under expanded condition with an underexpansion level of about 11.28% corresponding to NPR 7. It can be observed that the difference in the peak pressure between profile at X/D = 1 and X/D = 2 is quite large compared to that at NPR 6. Furthermore, the peak pressure at X/D = 2, 3, 5 and 8 are found to be fairly lower than that at NPR 6 for the corresponding X/D values suggesting a rise in Mach number with a corresponding rise in velocity at the nozzle end, owing to the favorable pressure gradient, while at X/D = 10, the peak pressure at NPR 7 dominates the peak pressure of NPR 6.

A closer look at Figure 23a reveals that the peak pressure for y profile at X/D = 2 and 3 is higher than that at NPR 6 signifying greater inertia of the jet. Meanwhile, the jet spread is also observed to be increased for the underexpanded case as can be substantiated by the spread of y profile at X/D = 1 and 2, respectively.

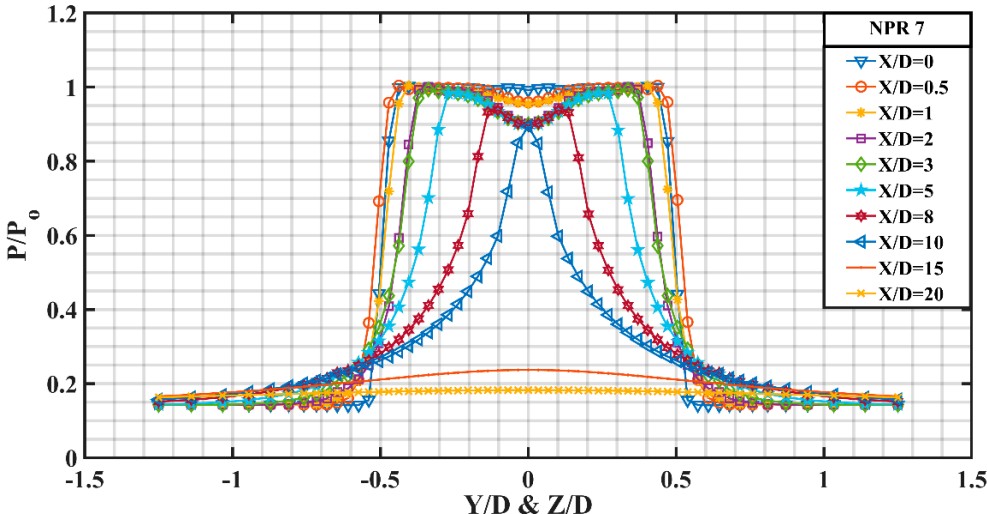

**Figure 22.** Pressure profile variation for uncontrolled jet at NPR 7.

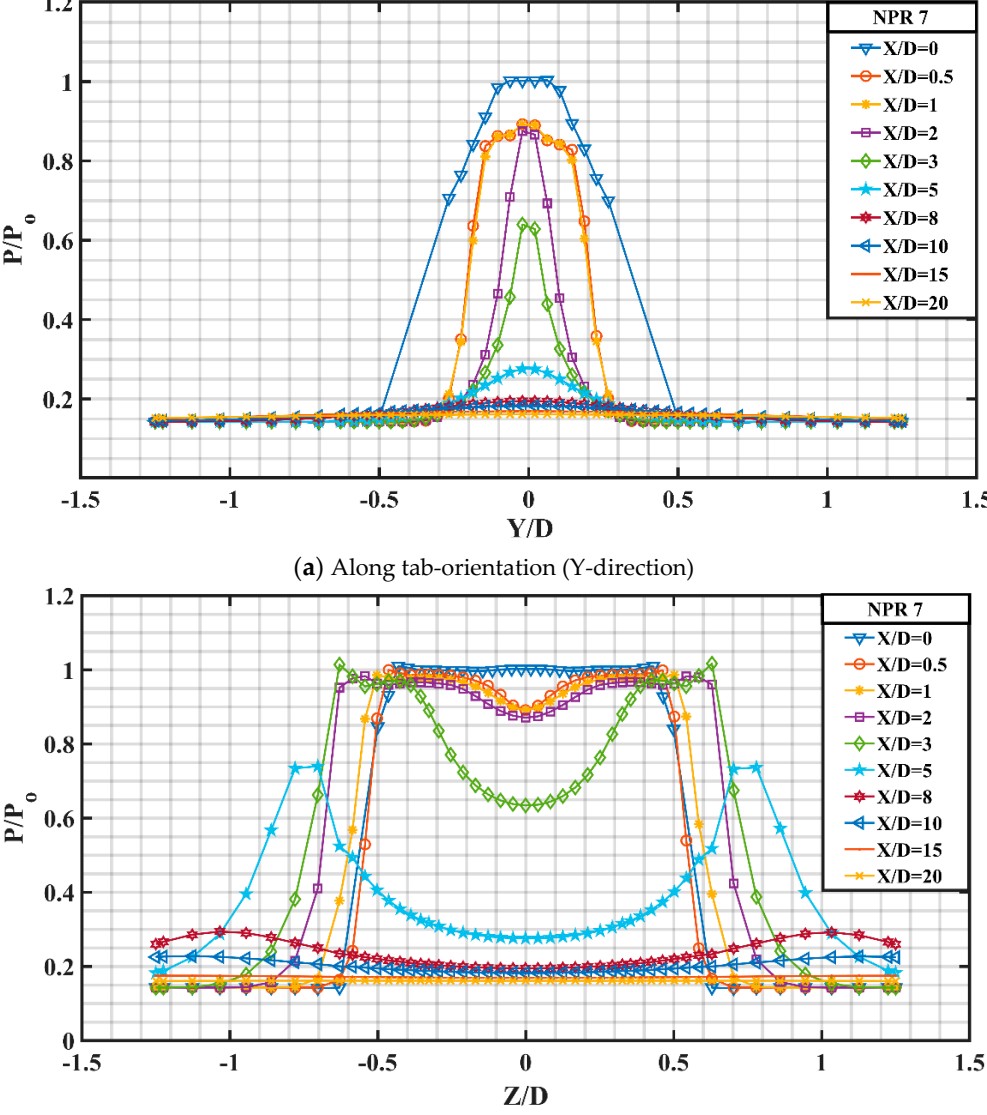

(**a**) Along tab-orientation (Y-direction)

(**b**) Perpendicular to tab-orientation (Z-direction)

**Figure 23.** Pressure profile variation for controlled jet at NPR 7.

The pressure level at X/D = 0, 0.5 and 1 is almost identical to NPR 6 except that the fairly invariant Mach zone at X/D = 0.5 and 1 around the jet at NPR 7 is consequentially thicker than that at NPR 6. There is a slightest hint of asymmetry which is somewhat visible at this underexpanded condition.

Figure 23b shows the z direction pressure profile for the controlled jet at NPR 7 and it can be observed that there is a marginal pressure difference between pressure profile at X/D = 1 and 2 compared to that at NPR 6. Furthermore, the dip at X/D = 2 and 3 is more for NPR 6 than for NPR 7. At X/D = 3, the two off-centered peaks show a constant pressure zone from Z/D = 0.4 to 0.6 and then show a gradual increase till Z/D = 0.7 and finally falls abruptly to attain a pressure ratio of $P/P_0$ = 0.18.

### 3.3. Mach Contours

In the present investigation, Mach contours at different NPRs for both uncontrolled and controlled jet helps to gauge the effectuality of rectangular tabs in inducing augmented blending to the jet flow.

Figure 24a depicts the Mach contour of uncontrolled jet corresponding to NPR 4, and from the contour, six prominent shock cells of abating strength are observed. Meanwhile, Figure 24b,c show the Mach contours for controlled jet at the same NPR in two different view planes; that is, when viewed perpendicular to tab-orientation (x-y plane) and when viewed along tab-orientation (x-z plane). Mach contour along x-y plane shows considerable reduction in the shock cells compared to the uncontrolled jet. Furthermore, Mach contour along the x-z plane shows the jet bifurcation and two high-speed fields on both sides of the jet axis can be indisputably observed. From the literature, it has been well established that varying the discharge conditions at the exit of the nozzle results in varying far field flows and these flows each have a peculiar characteristic that is predominately governed by its momentum flux. Plume bifurcation happens to be one such far field flow that is developed by proper blending of regulated circumferential and axial excitation at the nozzle end. A correct proportion of these excitation can lead to the separation of a single jet into two individual jets, such that each bifurcated jet holds half of the axial momentum [25,26]. In the present study, the excitation to the jet was offered by the short rectangular tabs at the nozzle exit [27].

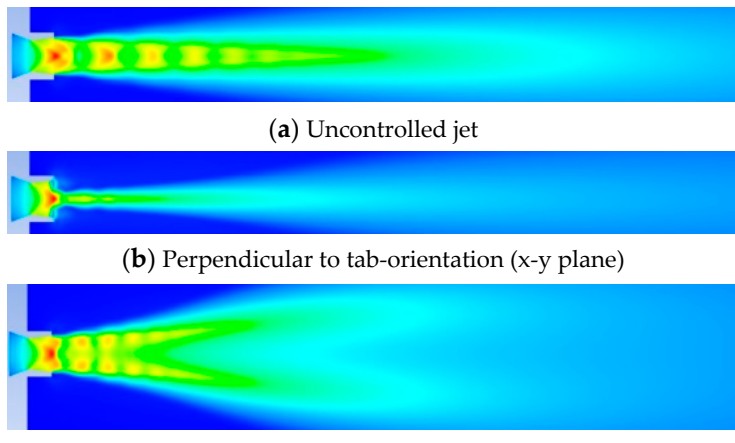

(**a**) Uncontrolled jet

(**b**) Perpendicular to tab-orientation (x-y plane)

(**c**) Along tab-orientation (x-z plane)

**Figure 24.** Mach contours of uncontrolled and controlled jets at NPR 4 (overexpansion).

Figures 25a, 26a, 27a and 28a show the Mach contours for uncontrolled jet at NPRs 5, 6, 7 and 8, respectively. It can be confirmed that the number of shocks, shock strength and shock cell length increase as the NPR is increased. Furthermore, Figures 25b,c, 26b,c, 27b,c and 28b,c show the Mach contours for controlled jet at NPRs 5, 6, 7 and 8 in x-y and x-z planes separately. The jet spread can be observed to increase with the increasing NPR values along with the increase in shock cell length and its strength.

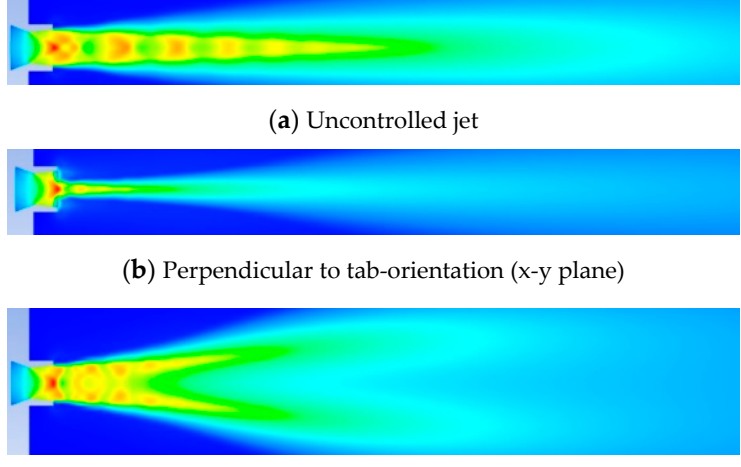

(**a**) Uncontrolled jet

(**b**) Perpendicular to tab-orientation (x-y plane)

(**c**) Along tab-orientation (x-z plane)

**Figure 25.** Mach contours of uncontrolled and controlled jets at NPR 5 (overexpansion).

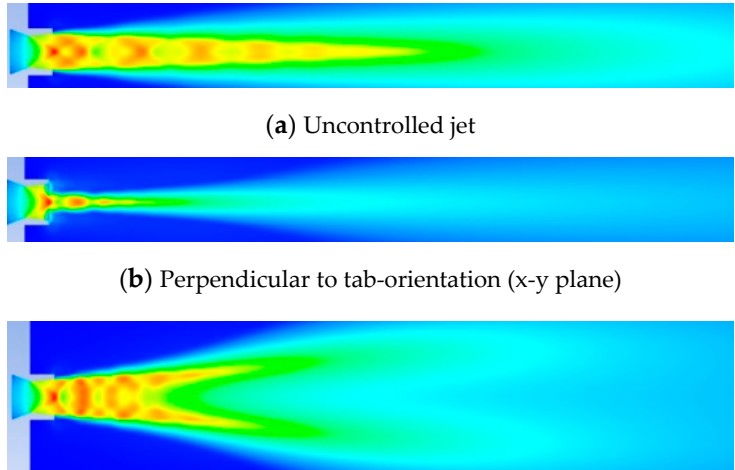

(**a**) Uncontrolled jet

(**b**) Perpendicular to tab-orientation (x-y plane)

(**c**) Along tab-orientation (x-z plane)

**Figure 26.** Mach contours of uncontrolled and controlled jets at NPR 6 (marginal underexpansion).

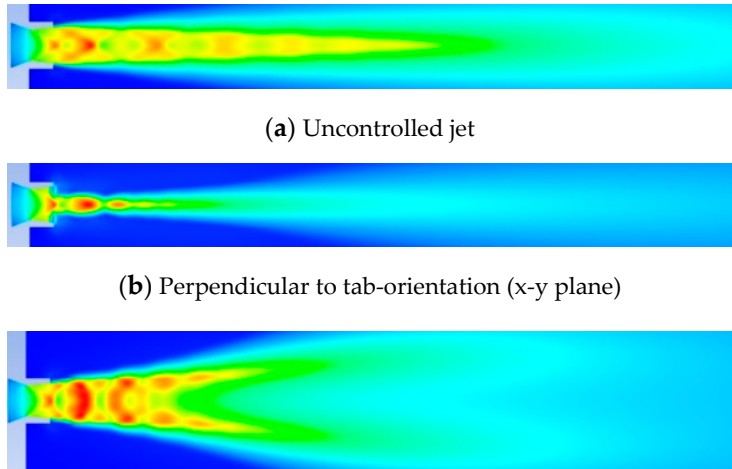

(**a**) Uncontrolled jet

(**b**) Perpendicular to tab-orientation (x-y plane)

(**c**) Along tab-orientation (x-z plane)

**Figure 27.** Mach contours of uncontrolled and controlled jets at NPR 7 (underexpansion).

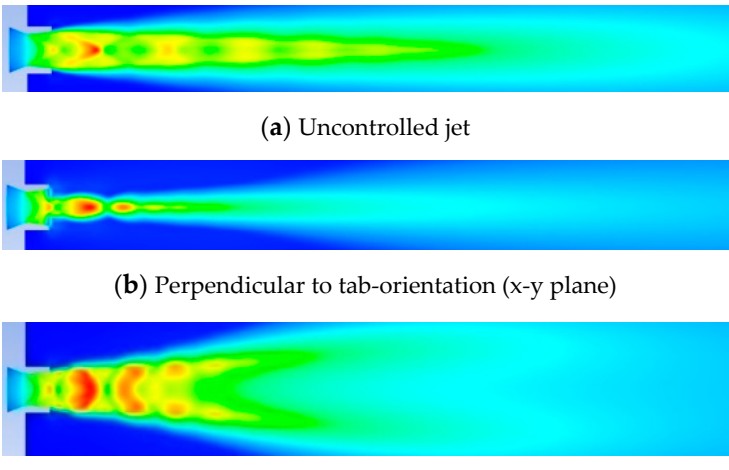

(**a**) Uncontrolled jet

(**b**) Perpendicular to tab-orientation (x-y plane)

(**c**) Along tab-orientation (x-z plane)

**Figure 28.** Mach contours of uncontrolled and controlled jets at NPR 8 (underexpansion).

### 3.4. Numerical Schlieren

Schlieren photography is a widely used ocular technique that optically visualizes the flow of fluids with varying density. In this present investigation, the attempt is to numerically replicate the same effect as that of optical schlieren so that the expansion fans, oblique shocks, barrel shocks and reflected compression waves are envisaged. Figures 29a, 30a, 31a, 32a and 33a depict the numerical schlieren for the uncontrolled jet at NPRs 4, 5, 6, 7 and 8, respectively, and a substantial number of barrel shock structures are formed in the core of the jet. While Figures 29b,c, 30b,c, 31b,c, 32b,c, 33b,c, presents the numerical schlieren for controlled jet at NPRs 4, 5, 6, 7 and 8 in x-y and x-z planes, respectively. Abatement in the length of shock cell structures for the controlled jet along with jet bifurcation and jet spread can be observed.

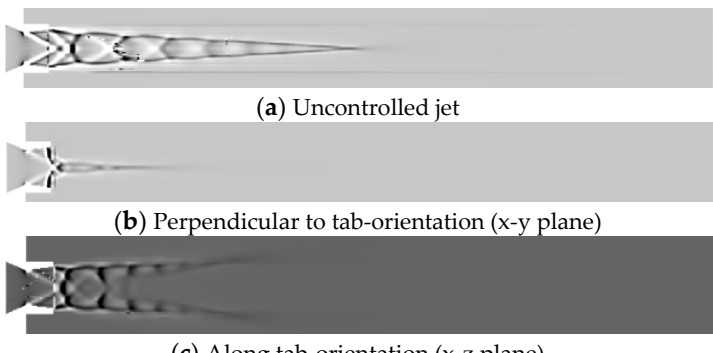

(**a**) Uncontrolled jet

(**b**) Perpendicular to tab-orientation (x-y plane)

(**c**) Along tab-orientation (x-z plane)

**Figure 29.** Schlieren views of uncontrolled and controlled jets at NPR 4 (underexpansion).

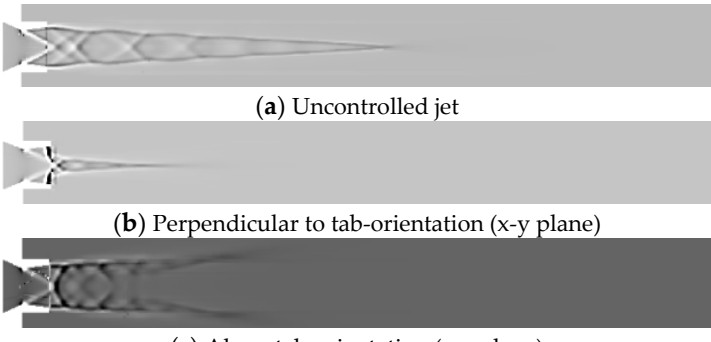

(**a**) Uncontrolled jet

(**b**) Perpendicular to tab-orientation (x-y plane)

(**c**) Along tab-orientation (x-z plane)

**Figure 30.** Schlieren views of uncontrolled and controlled jets at NPR 5 (underexpansion).

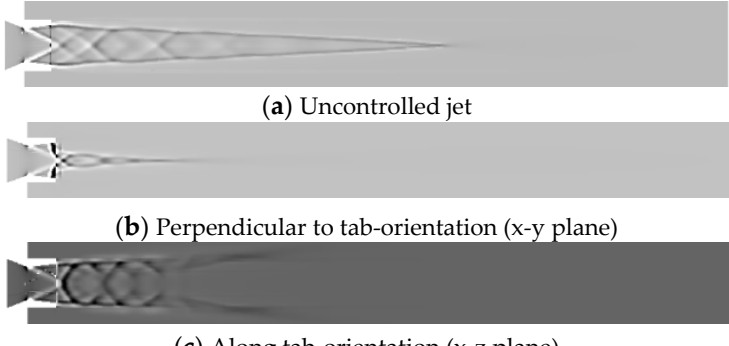

(**a**) Uncontrolled jet

(**b**) Perpendicular to tab-orientation (x-y plane)

(**c**) Along tab-orientation (x-z plane)

**Figure 31.** Schlieren views of uncontrolled and controlled jets at NPR 6 (underexpansion).

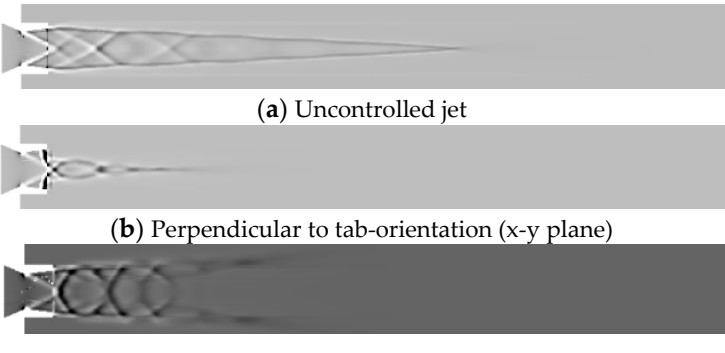

(**a**) Uncontrolled jet

(**b**) Perpendicular to tab-orientation (x-y plane)

(**c**) Along tab-orientation (x-z plane)

**Figure 32.** Schlieren views of uncontrolled and controlled jets at NPR 7 (underexpansion).

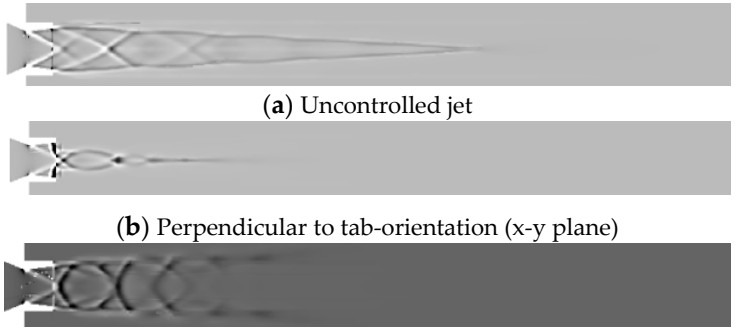

(**a**) Uncontrolled jet

(**b**) Perpendicular to tab-orientation (x-y plane)

(**c**) Along tab-orientation (x-z plane)

**Figure 33.** Schlieren views of uncontrolled and controlled jets at NPR 8 (underexpansion).

## 4. Conclusions

The present numerical investigation is aimed at stimulating the computational model of supersonic jet flow to imitate the mixing behavior in overexpanded, correctly expanded and under expanded conditions, respectively. The NPRs of 4, 5, 6, 7 and 8 are chosen to modestly cover all the expansion levels. The results obtained are independent of grid size above 2.7 million, and accordingly, it is chosen for further study to achieve computational economy. It is observed that prominent shock cell count increases with the escalation in NPR and the number count of shock cell structures for the controlled jet are less, compared with the uncontrolled jet at any investigated NPR. The core length for both controlled and uncontrolled jet actuators escalate with the escalating NPR owing to the abating adverse pressure gradient at the exit of the nozzle. Moreover, the percentage core length reduction first increases from NPR 4 to NPR 5 and then abates with an escalation in NPR, which indicates that the rectangular tabs perform better for underexpanded level and as much as 75% abatement in jet core length is attained with rectangular short tabs at NPR 5. Furthermore, the effect of tab increases the

amount of jet spread, in contrast to the uncontrolled jet. It can be concluded from the results that the jet bifurcates into two high-speed regions on both sides of the jet axis, away from the nozzle end owing to tabs. The Mach contour and the numerical schlieren results serve as qualitative support to this investigation and they show the number of shock cell structures, barrel shock, abatement in the jet core length and jet bifurcation visually.

**Author Contributions:** All the authors have contributed their efforts to complete the paper. A.R. performed numerical simulations and analyzed the results; M.K. supervised the work and reviewed and edited the manuscript; D.D. wrote the first draft; V.M. and M.U. co-wrote the first draft. All authors have read and agreed to the published version of the manuscript.

**Funding:** This research received no external funding.

**Acknowledgments:** The authors acknowledge that the Department of Mechanical Engineering at the National Institute of Technology, Patna, has provided computational resources to carry out the preliminary analysis. The authors deeply acknowledge the high-performance computing facility provided by the Department of Aerospace Engineering at the Indian Institute of Technology in Kharagpur, which has greatly contributed to this study. The assistance offered by Tamal Jana (a PhD student at the Department of Aerospace Engineering at the Indian Institute of Technology in Kharagpur) in the editing of the manuscript is deeply acknowledged.

**Conflicts of Interest:** The authors declare no conflict of interest.

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
