# Peer review of "Assessment of Short Rectangular-Tab Actuation of Supersonic Jet Mixing"

_actuators, doi:10.3390/act9030072_

Round 1

Reviewer 1 Report

The present work titled 'Assessment of Short Rectangular-Tab Actuation of Supersonic Jet Mixing' is a good effort towards the passive jet mixing. However, Authors need to resolve some of the critical issues before further proceed, and some of the issues are listed below as examples: 

1.  In abstract: The blockage ratio is specified 5. It is inconsistent with the information given in section 2. Justify. 

2. If we increase the length and decrease the width of the tab then how it will influence the flow. In other words, why did you chose this tab configuration? Please reflect on it. 

3. At line 122, the maximum skewness is mentioned 0.48, and at line 203 it is mentioned that the overall skewness comes down to a minimum of 0.50. Which one should be the correct skewness? Justify. 

4. Please mention the y+ value for the near-wall mesh used in this study. 

5. Since the flow conditions are not the same, please mentioned whether it is justified to compare the computational results with experimental data. 

6. The results of the reduction in core length are presented in Table 1 and Fig. 16. The authors should avoid repetition.  

7. Why the results of the jet structure, such as the influence of NPRs on overexpansion through underexpansion, presented in Mach contours and numerical Schlieren are not consistent. Please justify.      

8. The authors should avoid the use of abbreviated terms only in a new section. Please use the full form rather than the abbreviated form when first use in a new section; it will improve the readability of the manuscript. 

9. Some simple editing mistakes, such as unnecessary use of capital letters in a sentence should be corrected. 

10. The presentation of results is somewhere long and in-vogue. The results should be in a brief manner by avoiding repetition.   

Reviewer 2 Report

Review of actuators-891038 - Assessment of Short Rectangular-Tab Actuation of Supersonic Jet Mixing

The work deals with the mixing properties of a supersonic gas jet controlled with two short rectangular actuators. The work is well written but I think that the authors should address the two following major points:

A domain independence study have to be carried out.

The validation study it is not satisfactory: in order to validate the computational model the two jets must have the same conditions.

Reviewer 3 Report

This paper describes a sound full work well written and presented.

I have only one small comments why the authors write a force term in their momentum equations ? 

Reviewer 4 Report

Assessment of Short Rectangular-Tab Actuation of Supersonic Jet Mixing

  1. Numerical methodology is a key concept of this paper. The authors should hence demonstrate that the critical inferences on mixing, potential core changes, and shock cell structures, are invariant to the turbulence model used. The results should be repeatable with a different model as well.
  2. Figures 1 and 2 - Markings are not clear.
  3. Figures 3 and 4 - Quantify properties of near-wall grid spacing in terms of wall units.
  4. Include jet half-width plots for all NPRs tested.
  5. What is the physical mechanism leading to plume bifurcation in controlled jets?
  6. Does the authors observe axis switching? If not, remove Figure 29 and its description. This is a well-documented fact.

Round 2

Reviewer 1 Report

The authors are thanked for reviewing their manuscript. The authors mentioned in their review response that " the results in an increase in strength and number of the shock cell structures which can be clearly seen in Mach contours and Numerical schlieren images. Since the shock cell strength is varying with the nozzle pressure ratios, this leads to an inconsistency in the results of jet structure."  However, at a given NPR the jet structure such as shock cell length should be same in their corresponding Mach contours and Numerical Schlieren images. Otherwise it implies the presence of an error. Please reflect on it and justify.

Reviewer 2 Report

I still think that the validation it is not satisfactory. The differences between the two profiles in figure 10 are too big even if the behavior is the same.

I still think that the authors should use the same condtions for the two jets in order to properly validate their model.

Reviewer 4 Report

No further comments

Round 3

Reviewer 2 Report

The authors have addressed all my comments. Therefore I suggest to accepet the paper in the present form.